# Insights into the Underlying Mechanism of Ochratoxin A Production in *Aspergillus niger* CBS 513.88 Using Different Carbon Sources

**DOI:** 10.3390/toxins14080551

**Published:** 2022-08-12

**Authors:** Shan Wei, Chaojiang Hu, Ping Nie, Huanchen Zhai, Shuaibing Zhang, Na Li, Yangyong Lv, Yuansen Hu

**Affiliations:** 1College of Bioengineering, Henan University of Technology, Zhengzhou 450001, China; 2Henan Provincial Key Laboratory of Biological Processing and Nutritional Function of Wheat, Zhengzhou 450001, China

**Keywords:** *Aspergillus niger*, ochratoxin A, secondary metabolism, redox homeostasis, carbon sources, *AnGal4*

## Abstract

*Aspergillus niger* produces carcinogenic ochratoxin A (OTA), a serious food safety and human health concern. Here, the ability of *A. niger* CBS 513.88 to produce OTA using different carbon sources was investigated and the underlying regulatory mechanism was elucidated. The results indicated that 6% sucrose, glucose, and arabinose could trigger OTA biosynthesis and that 1586 differentially expressed genes (DEGs) overlapped compared to a non-inducing nutritional source, peptone. The genes that participated in OTA and its precursor phenylalanine biosynthesis, including *pks*, *p450*, *nrps*, *hal*, and *bzip*, were up-regulated, while the genes involved in oxidant detoxification, such as *cat* and *pod*, were down-regulated. Correspondingly, the activities of catalase and peroxidase were also decreased. Notably, the novel *Gal4*-like transcription factor An12g00840 (*AnGal4*), which is vital in regulating OTA biosynthesis, was identified. Deletion of *AnGal4* elevated the OTA yields by 47.65%, 54.60%, and 309.23% using sucrose, glucose, and arabinose as carbon sources, respectively. Additionally, deletion of *AnGal4* increased the superoxide anion and H_2_O_2_ contents, as well as the sensitivity to H_2_O_2_, using the three carbon sources. These results suggest that these three carbon sources repressed *AnGal4*, leading to the up-regulation of the OTA biosynthetic genes and alteration of cellular redox homeostasis, ultimately triggering OTA biosynthesis in *A. niger*.

## 1. Introduction

*Aspergillus* and *Penicillium* frequently contaminate nuts and grapes and produce the “group 2B carcinogen” ochratoxin A (OTA), which is a serious threat to human health and causes huge economic losses worldwide [1,2]. In *Aspergillus*, OTA is dominantly produced by *Aspergillus carbonarius* and *A**. ochraceus*, and recent studies reported that several *A.*
*niger* strains produced OTA [3,4]. *A. niger* is an extremely well-studied and versatile organism, with widespread applications in the production of organic acids, industrial enzymes, and heterologous proteins. *A. niger* CBS 513.88 was found to have a putative OTA biosynthetic gene cluster by genome sequencing [5]. Further investigation of the OTA biosynthesis in *A. niger* CBS 513.88 could, therefore, provide the theoretical basis for the development of effective strategies to control mycotoxin contamination in the fermentation of industrial products.

OTA is a polyketide derivative formed by connecting a dihydrocoumarin moiety and L-phenylalanine by an amide bond [6,7]. The OTA biosynthetic pathway has been well characterized in *A. ochraceus* using comparative genomic analyses [3]. Four genes, *pks*, *p450*, *nrps*, *hal*, and the *bzip* gene for a transcription factor (TF), comprise this cluster [8]. Previous reports indicated that OTA biosynthesis is regulated by environmental factors, including carbon sources, nitrogen sources, temperature, water activities, pH, and other conditions [9,10].

The utilization of carbon sources affects fungal production of secondary metabolites, including mycotoxins such as aflatoxin [11], fumonisin [12], patulin [13] trichothecene [14], citrinin [15], and OTA [16]. However, apart from carbon catabolite repression (CCR) [17], the cAMP signaling pathway [18], and the flux rearrangement of primary metabolism [19], few studies have evaluated the regulatory mechanisms of secondary metabolite production. Therefore, the underlying mechanism of carbon sources regulating OTA production in *A. niger* needs to be further explored. Glucose, sucrose, and arabinose could be used as carbon sources in industrial fermentation of *A. niger* [20,21,22]. Elucidating the regulatory mechanism of OTA biosynthesis in response to different carbon sources will help in developing effective strategies for controlling OTA contamination.

In this study, to better understand the OTA production facilitating its prevention and control in *A. niger*, the effects of sucrose, glucose, and arabinose on growth, conidia formation, and OTA biosynthesis in *A. niger* CBS 513.88 were investigated, and its underlying mechanism was elucidated by transcriptome analyses. Notably, a significantly down-regulated fungal-specific *AnGal4*-like Zn(II)2Cys6 transcription factor in three OTA-inducing carbon sources was investigated, and its function in OTA biosynthesis and underlying regulatory mechanisms were further elucidated.

## 2. Results

### 2.1. Identification of the OTA-Producing Aptitude by A. niger CBS 513.88

To investigate whether the *A. niger* CBS 513.88 strain could produce OTA, the retention time of the OTA standard was determined by HPLC-FLD (Figure 1A). The corresponding peak was observed at a similar time in the methanolic extracts of *A. niger* CBS 513.88 that was cultured on YES solid media under the same chromatographic conditions (Figure 1B), so the peak was further analyzed using UHPLC-MS. The results confirmed that *A. niger* CBS 513.88 could produce OTA (Figure 1C,D).

### 2.2. Influence of Different Carbon Sources on Colony Morphology, Conidia Production, and OTA Biosynthesis of A. niger CBS 513.88

After 5.5 days of culture, colony morphology and sporulation ability were significantly changed in *A. niger* CBS 513.88 when grown on yeast extract containing different carbon sources (sucrose, glucose, and arabinose), when compared with peptone. As shown in Figure 2A, on YEP solid medium, colonies were covered by dense dark brown conidial heads in the center. In contrast, on YES, YEG, and YEA solid media, colonies were light yellow. The conidia productions on YES, YEG, and YEA solid media were significantly lower than that on YEP solid medium (Figure 2B). Moreover, the colony diameters on YES, YEG, and YEA solid media were larger than that on YEP solid medium (Figure 2C). Together, the results indicated that the growth rates of the strains grown on YES, YEG, and YEA solid media were higher than that on YEP solid medium.

The influence of different carbon sources on OTA biosynthesis was further studied by HPLC-FLD. After 5.5 days of cultivation at 23 °C in the dark, the concentration of OTA reached around 4.19, 3.64, and 21.93 μg/g dry hyphae on YES, YEG, and YEA solid media, respectively. However, OTA was not detected on the YEP solid medium (Figure 2D). These results confirmed that arabinose was the best carbon source to produce OTA in *A. niger* CBS 513.88, followed by sucrose and glucose, while peptone inhibited OTA production. Arabinose, sucrose, and glucose were, therefore, defined as OTA-inducing carbon sources, and peptone as the non-inducing nutritional source.

### 2.3. Transcriptional Profiles and DEGs Analyses of A. niger CBS 513.88 Using Different Carbon Sources

To further reveal the mechanism of OTA production using different carbon sources, we performed transcriptome sequencing of the *A. niger* strain. Strains grown on YES, YEG, and YEA solid media were the test group, and YEP medium was the control group. We identified DEGs, which overlapped with the three OTA-inducing carbon sources and the non-inducing nutritional source, to narrow the scope of our studies. The samples were all analyzed at 5.5 days of incubation at 23 °C in the dark. Transcriptome analyses showed that 1778, 1618, and 1792 genes were significantly up-regulated on YES, YEG, and YEA solid media, respectively, whereas 1317, 1079, and 1579 genes were significantly down-regulated, respectively (Figure 3A). A total of 1586 DEGs overlapped on YES, YEG, and YEA solid media, of which 986 were up-regulated and 595 were down-regulated (Figure 3B). Among them, genes associated with the OTA precursor phenylalanine (An08g06800, An14g06010, and An15g02460) and OTA biosynthesis (*nrps*, *P450*, *hal*, and *bzip*) were significantly up-regulated (Table 1), which was confirmed by qRT-PCR analyses (Appendix A). Although the transcriptional level of *pks* was unchanged on YES and YEG media, it was significantly up-regulated on YEA medium with the highest OTA concentration. Together, the results showed that carbon sources affected the OTA production of *A.*
*niger* by controlling the levels of OTA and its precursor phenylalanine biosynthesis.

KEGG pathway and GO analyses for the 1586 DEGs were also conducted. According to the KEGG pathway analyses, DEGs were involved in carbohydrate metabolism, containing glycolysis/gluconeogenesis, and starch/sucrose/galactose/amino acids metabolism (Figure 3C). Genes involved in phenylalanine, histidine, tyrosine, tryptophan, and arginine biosynthesis were up-regulated (Table 1). According to GO terms analyses, DEGs were primarily involved in oxidative stress. In the molecular functions and biological processes category, DEGs were mainly associated with oxidoreductase activity and oxidation-reduction processes, respectively (Figure 3D).

### 2.4. Carbon Sources Affect the Cellular Redox Homeostasis of A. niger CBS 513.88

The carbon source affects the cellular redox status in filamentous fungi, and OTA production is closely related to oxidative stress [3,10,12]. The generated H_2_O_2_ can be catalyzed by CAT and POD. In this study, the expression levels of An08g08920 and An14g00690 that encode CAT and An09g00820 and An16g00630 that encode POD, were all significantly down-regulated in three OTA-inducing conditions, when compared to non-inducing conditions (Figure 4A) (Table 1). Correspondingly, the activities of the two antioxidant enzymes of *A. niger* strains were detected (Figure 4B). Compared to that on YEP solid medium, the activities of CAT on YES, YEG, and YEA solid medium decreased by 75.6%, 69.5%, and 62.8%, respectively. Similarly, compared to that on the YEP solid medium, the activities of POD on YES, YEG, and YEA solid medium decreased by 40.4%, 48.0%, and 64.6%, respectively. Together, the results indicated that carbon sources may affect OTA biosynthesis by altering the homeostasis of cellular redox.

### 2.5. Disruption of a Novel Gal4-like Transcription Factor Affects OTA Biosynthesis

Transcription factors (TFs) in gene regulatory networks have previously been reported to regulate intracellular metabolic processes in response to environmental changes [23]. There were 25 up-regulated TFs and 14 down-regulated TFs in the 3 OTA-inducing conditions (Appendix A). Among them, 20 TFs had homologous genes in yeast and were annotated according to the *Saccharomyces* Genome Database, SGD (https://www.yeastgenome.org/, accessed on 1 December 2021). Using SGD annotation, these TF genes were shown to be involved in carbon metabolism, stress response, amino acid catabolism, sterol biosynthesis, etc. An12g00840, a fungal-specific *Gal4*-like Zn(II)2Cys6 transcription factor (*AnGal4*), was significantly down-regulated in three OTA-inducing carbon sources. To characterize the role of *AnGal4* in OTA biosynthesis, an *AnGal4* deletion strain was constructed (Figure 5A). The samples were collected after 5.5 days at 23 °C, when grown in the three OTA-inducing conditions. The results showed OTA yields of the Δ*AnGal4* strain increased by 47.65%, 54.60%, and 309.23% on YES, YEG, and YEA solid media, respectively (Figure 5B). Correspondingly, the transcript levels of OTA biosynthetic genes of the Δ*AnGal4* strain were all obviously up-regulated in the presence of the three carbon sources (Figure 5C–E). Together, the results showed that *AnGal4* negatively regulated the OTA biosynthesis.

### 2.6. AnGal4 Affects Cellular Redox Homeostasis of A. niger

To further investigate the correlation between *AnGal4* and oxidative stress, the O^2−^ and H_2_O_2_ contents, and the sensitivity to H_2_O_2_ in the Δ*AnGal4* strain using three carbon sources were determined. The results showed that the O^2−^ contents in the Δ*AnGal4* strain increased by 158.37%, 80.57%, and 27.07% on YES, YEG, and YEA solid media, respectively (Figure 6A). The H_2_O_2_ contents in Δ*AnGal4* strain increased by 11.94%, 25.77%, and 20.41% on YES, YEG, and YEA solid media, respectively (Figure 6B). In addition, although the growth was unchanged in the Δ*AnGal4* and control strains under unstressed conditions, the Δ*AnGal4* strain exhibited significant growth defects when supplemented with 10, 20 and 30 mM H_2_O_2_, with the defects especially pronounced at high concentrations, showing that deletion of Δ*AnGal4* increased the sensitivity to H_2_O_2_ (Figure 6C). Together, the results suggested that *AnGal4* affected the cellular redox homeostasis of *A. niger* and that the deletion of Δ*AnGal4* increased cellular oxidative stress.

## 3. Discussion

*A. niger* CBS 513.88 is a “generally recognized as safe” industrial fermentation strain, and previous reports showed the presence of potential OTA biosynthesis genes in its genome [5]. We found that CBS 513.88 produced OTA, and confirmed that arabinose was the suitable carbon source for OTA production, followed by glucose and sucrose. Our results are consistent with those of Hashem et al. [16], who reported that glucose and sucrose were favorable carbon sources for OTA production, but inconsistent with those of Wang et al. [24], who reported that arabinose repressed OTA production in *A. ochraceus*. The differences might be attributable to the different physiological and genetic characteristics of the fungi [25]. In addition, we found that peptone suppressed OTA production. The inhibitory effect of peptone on mycotoxins may be attributed to the limited availability of malonyl-CoA or acetyl, because almost all of them were used for primary energy production rather than mycotoxin biosynthesis [26]. In *A. flavus*, YES medium induced aflatoxin biosynthesis, while YEP medium repressed it, consistent with OTA production in *A. niger*. Sequencing of the transcriptomes of the *A. niger* CBS 513.88 strain cultured in OTA-inducing conditions (YEA, YES, YEG) compared to non-inducing conditions (YEP) was performed. The common differences were investigated to elucidate the underlying regulatory mechanism.

The OTA biosynthetic pathway has been well characterized in *A. ochraceus* [3]. Four genes, *nrps*, *p450*, *hal*, and *bzip* in the OTA biosynthetic gene cluster, were all up-regulated when sucrose or glucose was used as the sole carbon source. More importantly, *pks*, the first and backbone gene in the OTA biosynthesis pathway, was significantly up-regulated with arabinose as the carbon source. The high expression of *pks* was an important reason for the highest OTA yield when arabinose was the carbon source in *A. niger.* The results indicated that carbon sources affected OTA biosynthesis by regulating the expression of OTA biosynthesis-associated genes in *A. niger*.

Medina et al. [25] and Abbas et al. [27] reported that amino acids, including phenylalanine, tyrosine, tryptophan, histidine, and arginine, induced OTA production. Phenylalanine, as the precursor for OTA biosynthesis, is sequentially synthesized via the shikimate and phenylalanine biosynthesis pathways [25]. Our results showed that An08g06800 (catalyzing steps two to six in the shikimate pathway), An14g06010 (catalyzing step eight of the phenylalanine biosynthesis), and An15g02460 (catalyzing step 10 of the phenylalanine biosynthesis), were all up-regulated in the presence of three OTA-inducing carbon sources. Additionally, the genes involved in tyrosine, tryptophan, histidine, and arginine biosynthesis were also up-regulated in the presence of three OTA-inducing carbon sources. Together, the results showed that carbon sources may affect OTA biosynthesis by regulating the biosynthesis of amino acids.

Oxidative stress is an important factor correlated with the biosynthesis of mycotoxins in fungi [1,28,29,30,31]. Reverberi et al. [32] suggested that oxidative stress agents, such as carbon tetrachloride or t-butyl hydroperoxide, induced OTA production in *A. ochraceus*, while antioxidants had the opposite effects [33]. In addition, oxidative stress can regulate OTA biosynthesis, which has been confirmed by the disruption of the oxidative stress-related genes, *Aoyap1* and *AoloxA* [33]. Our results showed that expression of the oxidant detoxification-related genes (*cat* and *pod*) and activities of the corresponding antioxidant enzymes (CAT and POD) on the three OTA-inducing carbon sources were lower overall than that on the non-inducing nutritional source. These results suggested that carbon sources may trigger OTA biosynthesis by altering cell redox homeostasis in *A. niger*, consistent with the observed association of mannose blocking fumonisin production with milder oxidative stress in *Fusarium proliferatum* [12].

A carbon source is a principal factor that regulates various metabolic processes via TFs [6,23]. We identified 39 significantly changed TFs that overlapped in the 3 OTA-inducing conditions (YEA, YES, YEG) compared to non-inducing conditions (YEP). These mainly included ZnCys, C_2_H_2_, NEG and the bZIP family (Appendix A). Among them, *BrlA*, a transcriptional activator for conidiation [34], was significantly down-regulated in the presence of the three OTA-inducing carbon sources. The findings indicate that the conidiation was inhibited. In addition, Zn(II)2Cys6 proteins are always involved in the biosynthesis of fungal secondary metabolites. For example, Zn(II)2Cys6 proteins *AflR*, *Fum21*, *GliZ*, *MclR*, *CtnA*, and *Gip2* are transcriptional activators for the production of the fungal secondary metabolites aflatoxin [35], fumonisin [36], gliotoxin [37], compactin [38], citrinin [39], aurofusarin [40], respectively. Our transcriptome analysis indicated that An12g00840, a fungal specific transcription factor with a *Gal4*-like Zn(II)2Cys6 binuclear cluster DNA-binding domain (*AnGal4*), was significantly down-regulated in the presence of three OTA-inducing carbon sources. Based on the above findings, we speculate that the *AnGal4* TF may be involved in the biosynthesis of OTA. Our results indicate that disruption of *AnGal4* significantly increased OTA yield, and the expressions of *pks*, *nrps*, *p450*, *bzip*, and *hal*, in three OTA-inducing carbon sources. Specifically, the OTA yields of the Δ*AnGal4* strain increased by 47.65%, 54.60%, and 309.23%, when sucrose, glucose, and arabinose were used as the sole carbon source, respectively. The results indicate that *AnGal4* activity mediated by the carbon source negatively regulated the OTA biosynthesis. There is only very limited information, including carbon catabolite repression (CCR) [17], cAMP signaling pathway [18], and the flux rearrangement of primary metabolism [19], concerning the involvement in the regulation of secondary metabolite production responding to different carbon sources. Among them, CCR genes *CreA* and *Cre1* regulate the production of several secondary metabolites, including cephalosporin C [41], lovastatin [42], aflatoxin [43], and OTA [24]. In our study, the novel TF *AnGal4* was identified for the first time and demonstrated to have a negative regulatory role in OTA biosynthesis responding to the different carbon sources in *A. niger*. In addition, a previous report suggested that *Gal4* of yeast bound to upstream activation sites of target genes with 5′-CGGN_5_(T/A)N_5_CCG-3′ in vivo, or (A/C)GGN_10–12_CCG in vitro [43]. The promoters of the OTA biosynthesis genes of *A. niger* were predicted using a dedicated promoter prediction website (https://services.healthtech.dtu.dk, accessed on 1 February 2022). Notably, we found that *AnGal4* had direct binding sites on the promoters of the OTA biosynthesis genes *pks*, *nrps*, *p450*, *bzip*, and *hal*, in vivo or in vitro (Appendix A). The results indicate that *AnGal4* may directly regulate OTA biosynthesis by binding to the OTA biosynthesis genes.

Previous reports indicated that the CCR gene *CreA* homologue gene *Mig1* is negatively correlated with the oxidative stress response [44]. We speculate that *AnGal4* regulated by the carbon source may be involved in the oxidative stress. Our results showed that deletion of *AnGal4* increased the contents of O_2_^−^ and H_2_O_2_, as well as the sensitivity to H_2_O_2_ in the presence of the three selected carbon sources, demonstrating that *AnGal4* alters the cellular redox homeostasis in *A. niger*. The Δ*AnGal4* strain had the highest O_2_^−^ and H_2_O_2_ concentrations in YEA medium, consistent with the highest OTA yield. The results suggest that carbon source-mediated *AnGal4* may regulate OTA biosynthesis by altering cellular redox homeostasis.

## 4. Conclusions

Our results confirm that arabinose, sucrose, and glucose contribute to OTA production in the industrial fermentation strain *A. niger* CBS 513.88, and characterize the novel *Gal4*-like TF An12g00840 (*AnGal4*). The results suggest that carbon sources repress *AnGal4*, leading to up-regulation of OTA biosynthetic genes and altered cellular redox homeostasis. These changes ultimately trigger OTA biosynthesis in *A. niger* CBS 513.88. This study clarifies the underlying regulatory mechanism of OTA biosynthesis responding to different carbon sources. The findings provide a theoretical basis for the development of effective strategies to control mycotoxin contamination in the fermentation of industrial products.

## 5. Materials and Methods

### 5.1. Construction of Strains and Cultivation Conditions

The strains and primers used in this study are listed in Table 2 and Appendix A, respectively. *A. niger* CBS 513.88 (provided by professor Li Pan, South China University of Technology) and *A. niger* CBS 513.88 (*kusA^−^*, *pyrG^−^*) (laboratory constructed) were grown on YEA (6% arabinose, 2% yeast extract, pH 5.8), YES (6% sucrose, 2% yeast extract, pH 5.8), YEG (6% glucose, 2% yeast extract, pH 5.8), and YEP (6% peptone, 2% yeast extract, pH 5.8) solid media that were not supplemented or supplemented with 10 mM uridine. In addition, the YEA, YES, YEG, and YEP solid media were also used for strain morphology and conidia assay, OTA production, enzyme activity assay, determinations of superoxide anion and hydrogen peroxide (H_2_O_2_) contents, and the H_2_O_2_ susceptibility assays. Czapek-Dox (CD) medium supplemented with or without uridine was used for strain transformation. All strains were cultivated at 30 °C for 5.5 days for growth and at 23 °C for 5.5 days in the dark for OTA production.

*AnGal4* (An12g00840) gene knockout was performed by homologous recombination [45], and *A. niger* CBS 513.88 (*kusA^−^*, *pyrG^+^*) was the control strain. The *pyrG* gene expression cassette was used as the selection marker, which was cloned from plasmid ANIp7. Homologous fragments of 1.60 kb and 1.47 kb at either end of the *AnGal4* open reading frame (ORF) were amplified. The upstream and downstream homologous fragments and *pyrG* expression cassette were ligated into plasmid pMD20 using a one-step cloning kit (Novazyme, Nanjing, China) to yield pMD20-*ΔAnGal4*, which was transferred into the *A. niger* CBS 513.88 (*kusA^−^*, *pyrG^−^*) strain. The recombinant strain was screened using Czapek-Dox (CD) medium lacking uridine. The screening was verified by PCR using the genome of the correct transformants as a template.

### 5.2. Colony Morphology, Mycelial Growth, and Conidia Production

Conidia suspensions (5 μL, 10^6^ spores/mL) were centrally spotted on YEA, YES, YEG, and YEP media and cultivated for 5.5 days at 23 °C. Colony morphology and colony diameters were observed and measured [46]. Conidia suspensions were viewed using a microscope and counted using a hemocytometer. All cultures were conducted in triplicate.

### 5.3. High-Performance Liquid Chromatography (HPLC) Analysis of OTA

Conidia suspensions (200 μL, 10^6^ spores/mL) were spread evenly on YEA, YES, YEG, and YEP media and cultivated for 5.5 days at 23 °C in the dark. All mycelium and agar on plates were collected, and then were mashed and placed in an Erlenmeyer flask with 25 mL methanol, followed by shaking for 1 h in the dark. All cultures were conducted in triplicate. OTA was detected by HPLC with fluorescence detection (HPLC-FLD) (Agilent Technologies, San Jose, CA, USA) [47].

### 5.4. Ultra-High Performance Liquid Chromatography Mass Spectrometry (UHPLC-MS) Analysis

For UHPLC-MS analysis, the supernatant of the OTA extract was purified using an OTA immunoaffinity column (PriboFast, Beijing, China) and transferred to a 2 mL Eppendorf tube. The highly purified sample was detected using a Q-Orbitrap instrument system (Thermo Fisher Scientific, Waltham, MA, USA), equipped with a binary solvent delivery system. A C18 column (2.1 × 100 mm, 1.7 µm particle diameter; Thermo Fisher Scientific) was used for chromatography. The UHPLC-MS run used positive ESI. The details were performed as previously described [48].

### 5.5. Transcriptome Analysis

Conidia suspensions (200 μL, 10^6^ spores/mL) were spread evenly on YEA, YES, YEG, and YEG media and cultivated for 5.5 days at 23 °C in the dark. Hyphae were collected, then frozen in liquid nitrogen and subjected to transcriptome analyses. We first extracted total RNA using a UNIQ-10 TRIzol RNA Purification Kit (Sangon Biotech, Shanghai, China), with three biological parallels for each sample, then sequenced it using a NovaSeq (Personalbio, Nanjing, China). Raw data showed significant differences between YES and YEP, YEG and YEP, and YEA and YEP, with a bioproject accession number of PRJNA765465. Significant differences were indicated by an absolute log_2_-fold change of 1.0 or greater, and p-values of 0.05 or less. All annotations were obtained according to NCBI (https://www.ncbi.nlm.nih.gov/, accessed on 10 December 2021). Gene Ontology (GO) terms and Kyoto Encyclopedia of Genes and Genomes (KEGG) pathways were performed according to the reference [49], and were classified as significantly enriched among the differentially expressed genes (DEGs).

### 5.6. Real-Time Quantitative PCR (RT-qPCR) Analysis

The qPCR data were analyzed according to a previously described method [50]. After RNA extraction, we used a PrimeScript™ RT reagent Kit (TaKaRa Bio, Beijing, China) to obtain the cDNA. We then used *qgpdA* as the reference gene, and the equation N = 2^−ΔΔCt^ to evaluate the relative transcription levels. The *t*-test was performed to assess the differences between samples.

### 5.7. Determination of Antioxidant Enzyme Activity, and Superoxide Anion and H_2_O_2_ Contents

Two hundred microliter aliquots of conidia suspensions containing 10^6^ spores/mL were spread evenly on YEA, YES, YEG, and YEP media and cultivated for 5.5 days at 23 °C in the dark. The mycelia were collected, triturated with acetone, and the supernatant was collected by centrifugation at 4 °C. Catalase (CAT) and glutathione peroxidase (POD) were quantified using the corresponding detection kits (Solarbio, Beijing, China). The superoxide anion (O^2−^) and H_2_O_2_ contents were also measured according to the corresponding detection kit instructions (Solarbio, Beijing, China).

### 5.8. Assay of H_2_O_2_ Susceptibility

Five microliter aliquots of conidia suspensions containing 10^6^ spores/mL were spotted onto YEA, YES, YEG, and YEP media supplemented with H_2_O_2_ at different concentrations (0, 10, 20, and 30 mM), and cultivated for 5.5 days at 30 °C. The growth of strains on plates was then recorded.

### 5.9. Statistical Analysis

All experiments were repeated with three independent biological replicates. The data are expressed as the mean ± standard deviation of independent tests performed in triplicate. The differences were calculated with student’s *t* test (*t*-test), and marked with *, **, and *** as *t* < 0.05, *t* < 0.01, and *t* < 0.001, respectively.

## Figures and Tables

**Figure 1 toxins-14-00551-f001:**
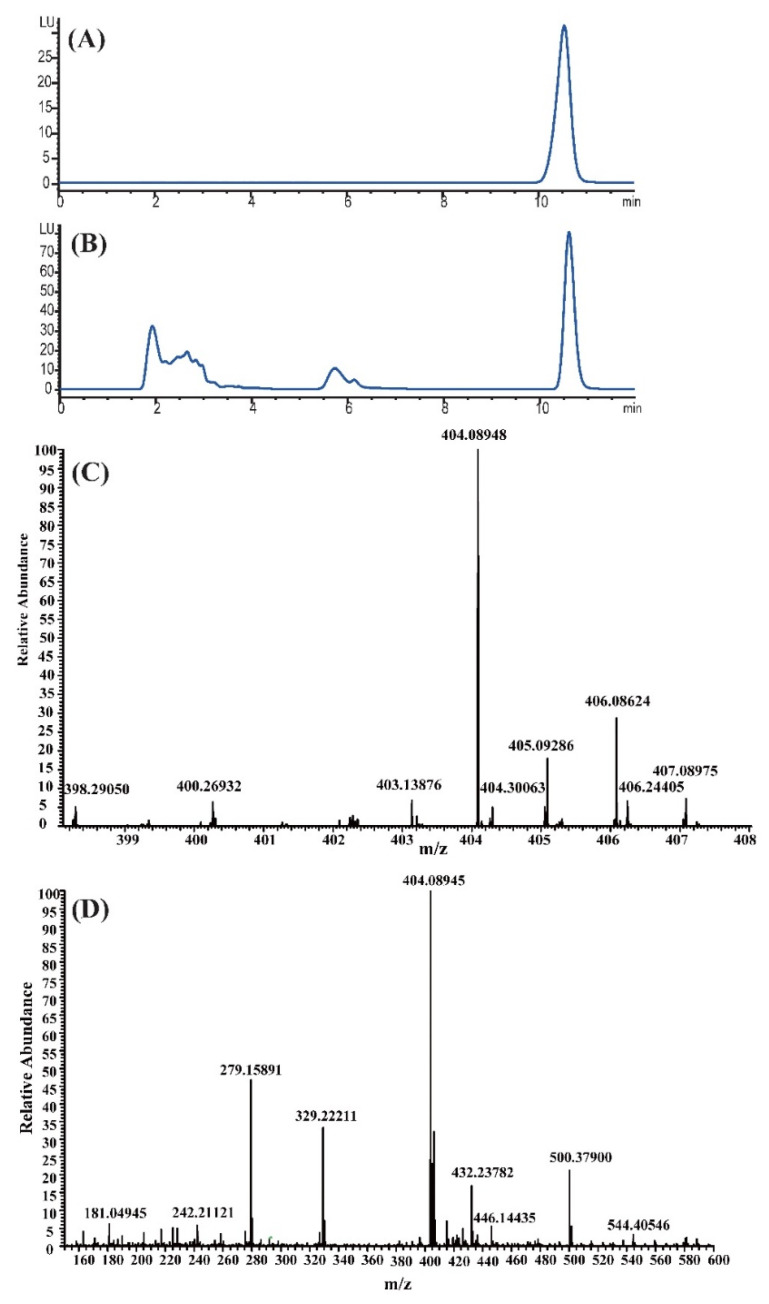
High-performance liquid chromatography with fluorescence detection (HPLC-FLD) chromatograms and ultra-high performance liquid chromatography mass spectrometry (UHPLC-MS) analysis of the methanolic extract of *A. niger* CBS 513.88. HPLC-FLD chromatograms of OTA standard (**A**) and the methanolic extract of *A. niger* CBS 513.88 (**B**), UHPLC-MS analysis of OTA standard (**C**) and the methanolic extract of *A. niger* CBS 513.88 (**D**) in the positive ion mode.

**Figure 2 toxins-14-00551-f002:**
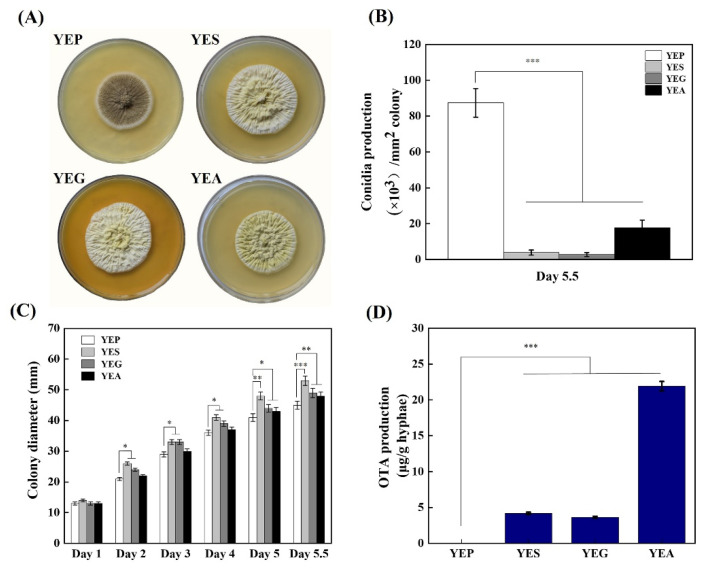
Colony view and phenotypic analysis of the *A. niger* CBS 513.88 strain inoculated on YEP, YES, YEG, and YEA media plates incubated at 30 °C for 5.5 days. (**A**) The colonies at 5.5 days. (**B**) Conidia production per mm^2^ of colony at 5.5 days. (**C**) The colony diameters at days 1~5.5. (**D**) OTA production (µg/g dry hyphae). The error bars represent the standard deviation of three parallel biological experiments. * *t* < 0.05, ** *t* < 0.01, *** *t* < 0.001.

**Figure 3 toxins-14-00551-f003:**
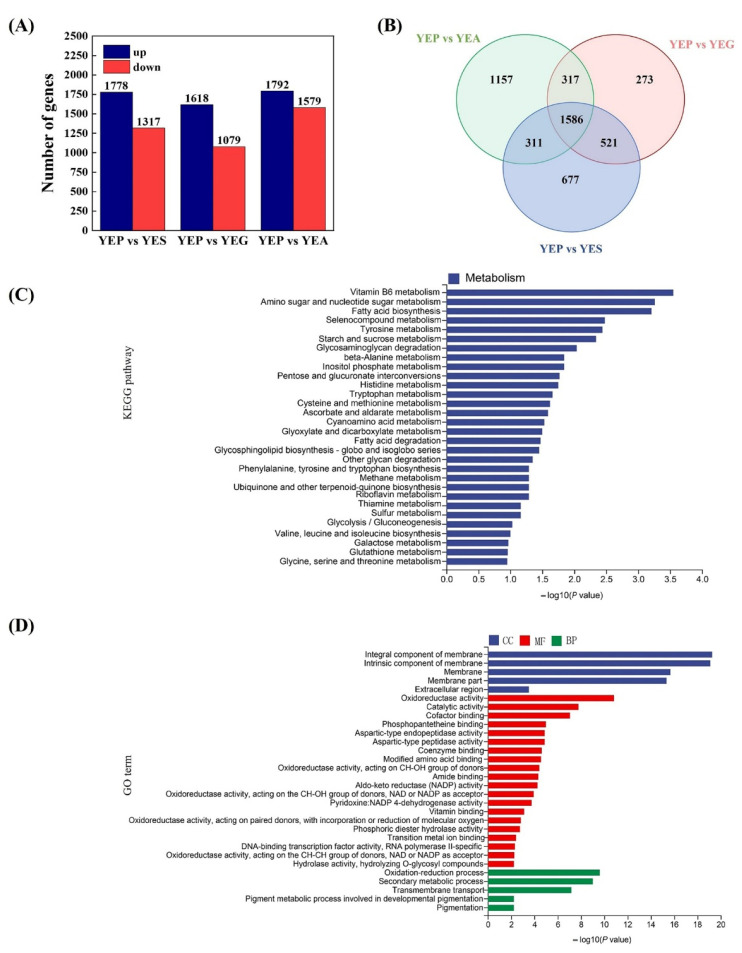
Analysis of differentially expressed genes in the presence of different carbon sources. (**A**) The number of up-and down-regulated genes on YES, YEG, and YEA solid media, when compared to that on YEP solid medium. (**B**) Venn diagram showing the overlaps on YEP vs. YES, YEP vs. YEG, and YEP vs. YEA. (**C**) Kyoto Encyclopedia of Genes and Genomes enrichment analysis of the overlapping differentially expressed genes (DEGs) on YES, YEG, and YEA solid media. (**D**) Gene Ontology enrichment analysis of the overlapping DEGs on YES, YEG, and YEA solid media.

**Figure 4 toxins-14-00551-f004:**
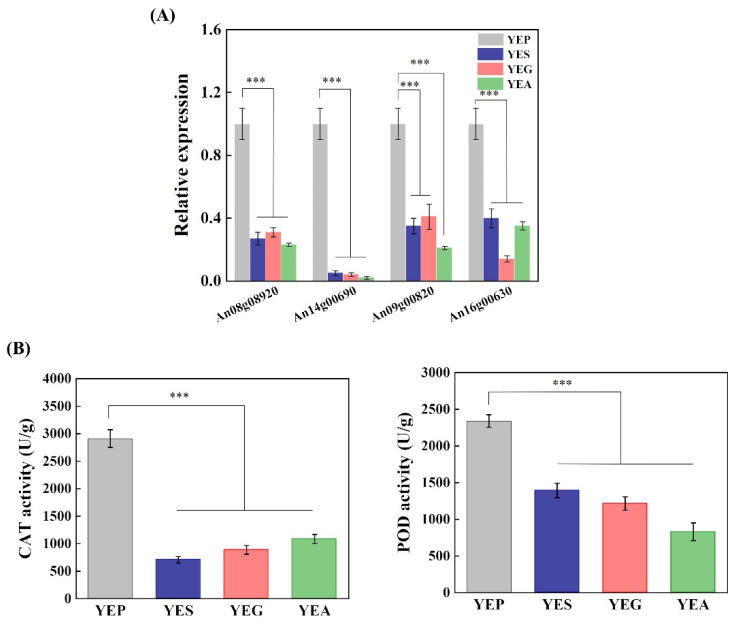
Effect of carbon sources on the cellular redox homeostasis of *A. niger* CBS 513.88. (**A**) The qRT-PCR results of differentially expressed genes associated with cellular oxidant detoxification. (**B**) The activities of antioxidant enzymes include catalase (CAT, U/g) and peroxidase (POD, U/g). All data are expressed as the mean ± standard deviation of independent tests performed in triplicate (*** *t* < 0.001).

**Figure 5 toxins-14-00551-f005:**
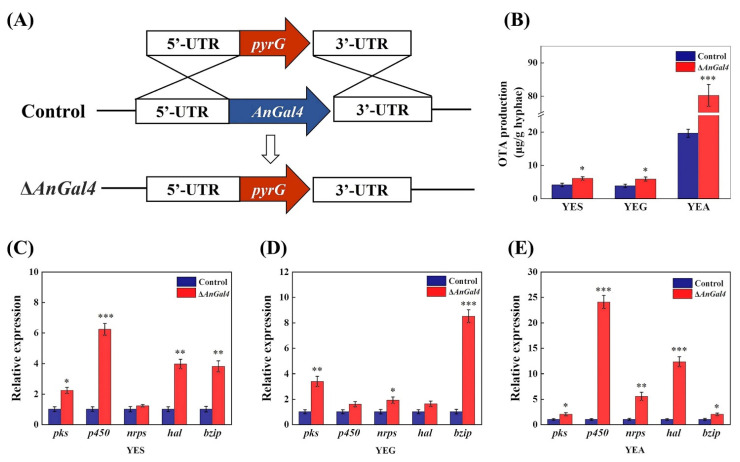
The effect of *AnGal4* deletion on the OTA yields and related gene expressions in the presence of three OTA-inducing carbon sources. Construction of the *AnGal4* deletion strain (**A**). The effect of *AnGal4* deletion on OTA yield (**B**). The effect of *AnGal4* deletion on the expressions of OTA biosynthetic genes on YES solid medium (**C**), on YEG solid medium (**D**), and on YEA solid medium (**E**). * *t* < 0.05, ** *t* < 0.01, *** *t* < 0.001.

**Figure 6 toxins-14-00551-f006:**
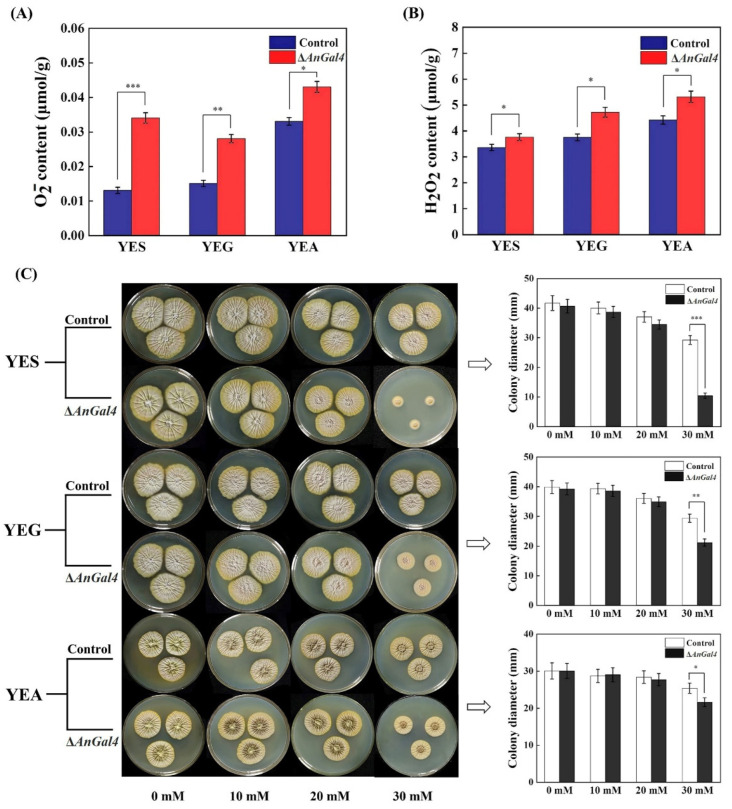
The effect of AnGal4 deletion on the cellular redox homeostasis of *A. niger* strain in the presence of three OTA-inducing carbon sources. (**A**) O^2−^ contents. (**B**) H_2_O_2_ contents. (**C**) The sensitivity to H_2_O_2_ at different concentrations and the effects of H_2_O_2_ on the growth of different strains. * *t* < 0.05, ** *t* < 0.01, *** *t* < 0.001.

**Table 1 toxins-14-00551-t001:** RNA-seq analysis of DEGs in *A. niger* CBS 513.88.

Gene ID	Function	Transcriptome Data (log_2_(Fold Change))		
		YES/YEP	YEG/YEP	YEA/YEP
**DEGs involved in conidiogenesis**				
An01g10540	Regulatory protein (*brlA*)	−2.44 ***	−2.18 ***	−2.81 ***
**DEGs involved in phenylalanine, histidine, tyrosine, tryptophan, and arginine biosynthesis**				
An08g06800	5-Dehydroshikimate dehydrase	1.30 ***	1.06 ***	1.71 ***
An14g06010	Chorismate mutase	1.28 ***	1.57 ***	1.90 ***
An15g02460	L-kynurenine/alpha-aminoadipate aminotransfease	1.45 **	1.19	1.50 ***
An14g07210	Histidinol dehydrogenase 1	3.37 ***	2.58 *	2.17 ***
An16g02500	Tryptophan synthase	4.33 ***	3.87 ***	3.40 ***
An04g08100	Shikimate/quinate 5-dehydrogenase	5.49 ***	5.66 ***	4.53 ***
An15g02340	Argininosuccinate synthase	1.61 ***	1.05 ***	1.16 ***
**DEGs involved in OTA biosynthesis**				
An15g07920	Polyketide synthase (*pks*)	0.01	0.19	1.7 ***
An15g07900	Oxidoreductase activity (*p450*)	3.28 **	2.62	2.30 **
An15g07910	Nonribosomal peptide synthase (*nrps*)	3.43 *	2.96 *	3.28 ***
An15g07880	Oxidoreductase activity (*hal*)	3.55 *	3.19	3.59 ***
An15g07890	bZIP transcription factor (*bzip*)	2.32	2.29	1.53
**DEGs involved in oxidative stress**				
An08g08920	Catalase activity (*cat*)	−1.42 ***	−1.41 ***	−2.15 ***
An14g00690	Catalase activity (*cat*)	−2.65 ***	−2.04 ***	−3.21 ***
An09g00820	Predicted peroxidase activity (*pod*)	−2.04 ***	−2.40 ***	−2.75 ***
An16g00630	Predicted peroxidase activity (*pod*)	−2.16 **	−3.67 ***	−1.59 *

All data are the mean ± standard deviation of independent tests performed in triplicate. * *p* < 0.05, ** *p* < 0.01, *** *p* < 0.001.

**Table 2 toxins-14-00551-t002:** *A. niger* strains used in this study.

*A. niger* Strains	Description	Sources
*A.niger* CBS 513.88	Wild-type strain	Provided by Li Pan
*A. niger* CBS 513.88 (*kusA^−^*, *pyrG^−^*)	*A. niger* CBS 513.88 derivative, Δ*kusA::ptrA*, Δ*pyrG::hygB*	Laboratory constructed
*A. niger* CBS 513.88 (*kusA^−^*, *pyrG^+^*) (Control)	*A. niger* CBS 513.88 derivative, Δ*kusA*, Δ*pyrG, pyrG-Com*	Laboratory constructed
Δ*AnGal4*	*A. niger* CBS 513.88 derivative, Δ*kusA*, Δ*pyrG*, Δ*AnGal4::pyrG*	This study

## Data Availability

Data are available in a publicly accessible repository.

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
