# Peer review of "Insights into the Underlying Mechanism of Ochratoxin A Production in Aspergillus niger CBS 513.88 Using Different Carbon Sources"

_toxins, 2022, doi:10.3390/toxins14080551_

Round 1
Reviewer 1 Report
This paper deals with the capacity by a strain of A. niger, a species employed in several industrial bioprocesses, to produce ochratoxin A in media with different carbon sources, along with the underlying regulatory mechanisms elucidated by means of transcriptome analysis. It is informative and quite well written, and in my opinion it can be accepted after minor revision as per the below list of corrections. Moreover, a careful re-reading is suggested to avoid redundant use of words throughout the manuscript.
Title of section 2.1: you do not identify the strain, but its OTA-producing aptitude; change to 'Identification of the OTA-producing aptitude by A. niger CBS 513.88'; lines 67-68: you did not perform HPLC analysis of the strain, but of its methanolic extract (according to what stated in section 5.3). Hence, change these lines to 'The corresponding peak was observed at a similar time in the methanolic extracts of the strain under the same chromatographic conditions'; caption of Fig.1: change to 'methanolic extract of A. niger...';
table 1: correct to 'dehydroshikimate dehydrase';
line 241: correct to 'cell';
line 244: correct 'nutrient source' to 'factor'; lines 250-251 : you mean 'Zn(II)2Cys6 proteins are always involved in the biosynthesis of fungal secondary metabolites'? line 262: correct to 'were' (subject is plural); line 268: maybe 'regulate production of several secondary metabolites,' sounds better; line 277: delete 'the' at the end of the line; line 281: correct to 'showed'; line 375: correct to 'was' (subject is singular).
Author Response
Response to Reviewer 1 Comments
Point 1: Lines 56-62: there is some redundancy in this last part of the introduction. I suggest to modify it as follows: 'In this study, the effects of sucrose, glucose, and arabinose on growth, conidia formation, and OTA biosynthesis in A. niger CBS 513.88 were investigated, and its underlying mechanisms were elucidated by transcriptome analyses. Notably, a significantly down-regulated fungal specific AnGal4-like Zn(II)2Cys6 transcription factor in three OTA-inducing carbon surces was investigated, and its function in OTA biosynthesis and underlying regulatory mechanisms were further elucidated.'; note that at line 58 (and again at line 166) 'downregulated' should be hyphenated to conform to the style used throughout the manuscript; the same to be done for 'upregulation' at lines 18, 25, 124 and 292.
Response 1: Thanks for your professional comments and good advice. In the last part of the introduction, we have modified the manuscript according to your suggestions, please find it in our revised manuscript (Line 57-63). Moreover, ‘downregulated’ was corrected to ‘down-regulated’ and 'upregulation' was corrected to ‘up-regulation’, we have checked the whole manuscript.
Introduction: In this study, to better understand the OTA production facilitating its prevention and control in A. niger, the effects of sucrose, glucose, and arabinose on growth, conidia formation, and OTA biosynthesis in A. niger CBS 513.88 were investigated, and its underlying mechanisms were elucidated by transcriptome analyses. Notably, a significantly down-regulated fungal specific AnGal4-like Zn(II)2Cys6 transcription factor in three OTA-inducing carbon sources was investigated, and its function in OTA biosynthesis and underlying regulatory mechanisms were further elucidated.
Point 2: Title of section 2.1: you do not identify the strain, but its OTA-producing aptitude; change to 'Identification of the OTA-producing aptitude by A. niger CBS 513.88'; lines 67-68: you did not perform HPLC analysis of the strain, but of its methanolic extract (according to what stated in section 5.3). Hence, change these lines to 'The corresponding peak was observed at a similar time in the methanolic extracts of the strain under the same chromatographic conditions'; caption of Fig.1: change to 'methanolic extract of A. niger...';
Response 2: Thanks for your professional comments. The title of section 2.1 has been revised according to your suggestions (Line 67). Title of section 2.1: Identification of the OTA-producing aptitude by A. niger CBS 513.88. The contents of section 2.1 has been revised in our new manuscript (Line 70-72), and the lines have been changed to 'The corresponding peak was observed at a similar time in the methanolic extracts of A. niger CBS 513.88 that was cultured on YES solid media under the same chromatographic conditions'. The caption of Figure 1 have been revised in our new manuscript.
Figure 1. High-performance liquid chromatography with fluorescence detection (HPLC-FLD) chromatograms and ultra-high performance liquid chromatography mass spectrometry (UHPLC-MS) analysis of the methanolic extract of A. niger CBS 513.88. HPLC-FLD chromatograms of OTA standard (A) and the methanolic extract of A. niger CBS 513.88 (B), UHPLC-MS analysis of OTA standard (C) and the methanolic extract of A. niger CBS 513.88 (D) in the positive ion mode.
Point 3: table 1: correct to 'dehydroshikimate dehydrase';
Response 3: Thanks for your advice. ‘dehydroshikimate dehdrase’ was corrected to ‘dehydroshikimate dehydrase’, see Table 1.
Point 4: line 241: correct to 'cell';
Response 4: Thanks for your advice. ‘cellar redox homeostasis’ was corrected to ‘cell redox homeostasis’, see line 255.
Point 5: line 244: correct 'nutrient source' to 'factor'; lines 250-251 : you mean 'Zn(II)2Cys6 proteins are always involved in the biosynthesis of fungal secondary metabolites'? line 262: correct to 'were' (subject is plural); line 268: maybe 'regulate production of several secondary metabolites,' sounds better; line 277: delete 'the' at the end of the line; line 281: correct to 'showed'; line 375: correct to 'was' (subject is singular).
Response 5: The correction has been made one by one in our revised manuscript. 'nutrient source' was corrected to 'factor' (Line 258); 'Zn(II)2Cys6 proteins are always involved as fungal secondary metabolites' was corrected to ' Zn(II)2Cys6 proteins are always involved in regulate production of several secondary metabolites' (Line 264-265); 'was' was corrected to 'were' (Line 276); 'regulate aspects of several secondary metabolite production' was corrected to 'regulate production of several secondary metabolites' (Line 282-283); 'showe' was corrected to 'showed' (Line 297); 'were' was corrected to 'was' (Line 392).
Reviewer 2 Report
Manuscript Number: toxins-1837911
Title: Insights into the underlying mechanism of ochratoxin A production in Aspergillus niger CBS 513.88 using different carbon sources
The submitted manuscript presents the results of the study on the effects of sucrose, glucose, and arabinose on growth, conidia formation, and OTA biosynthesis in A. niger. This topic is important from a practical point of view because A. niger grown on sugar-rich media is widely used for the biosynthesis of some organic acids and other compounds. In addition, the paper explains the basic regulatory mechanism of OTA biosynthesis responding to different carbon sources, which is of significant scientific importance. In my opinion, the submitted manuscript is a valuable extension of the existing knowledge base related to toxigenicity of A. niger and may be of interest to the readers of the journal. Overall, the paper is quite fairly written and organized. Below, there are some comments and suggestions:
lines 67: I suggest to specify more precisely what the peak in Fig. 1B refers to by adding the name of the species of mould and the substrate on which it was cultured e.g.: “The peak for A. niger CBS 513.88 that was cultured on …. was observed at…”
Line 85: The use of the word "longer" for the size of the diameter seems inappropriate.
Figure 2: There are no standard deviations in diagram (D). If the analysis was carried out in several repetitions, it is recommended to show them.
Table 5: The caption under table indicates that the standard deviation should be presented in the table.
Figure 3: The figure shows only 4 diagrams/fig. (A-D), whilst the caption suggests that there should be 5 (A-E).
Figure 5: There is no reference to figs (D) and (E) in the figure caption.
Figure 6. The figure shows only 3 diagrams/fig. (A-C), whilst the caption suggests that there should be 4 (A-D).
Materials and methods: It is advisable that the authors describe the methods used in the statistical analysis, the results of which are presented in the charts.
Editorial mistakes:
Line 98: “mm2” instead of “mm2”
Reviewer 3 Report
As general comment the work is well written and designed with relevant results.
In general terms the topic of the article is interesting, the methodology is explicitly presented and the results reported are interesting.
The structure of the paper is correct.
In my opinion, the abstract is too general, please reframe.
The introduction chapter should end with a paragraph indicating the purposefulness of the conducted research. Authors should clearly define the purpose of the work and formulate research hypotheses.
How was the statistical calculations done?
Materials and method section is well described and correspond to the aim set out in the manuscript. The table and figures clearly presenting the obtained results with their appropriate interpretation.
The references are sufficient and necessary.
The paper needs some editorial corrections.
I recommend the publication of this manuscript in the Toxins journal after minor revisions.
